# Improving GAN Training via Binarized Representation Entropy (BRE) Regularization

**Yanshuai Cao, Gavin Weiguang Ding, Kry Yik-Chau Lui, Ruitong Huang**
Borealis AI
Canada

## Abstract

We propose a novel regularizer to improve the training of Generative Adversarial Networks (GANs). The motivation is that when the discriminator $D$ spreads out its model capacity in the right way, the learning signals given to the generator $G$ are more informative and diverse. These in turn help $G$ to explore better and discover the real data manifold while avoiding large unstable jumps due to the erroneous extrapolation made by $D$. Our regularizer guides the rectifier discriminator $D$ to better allocate its model capacity, by encouraging the binary activation patterns on selected internal layers of $D$ to have a high joint entropy. Experimental results on both synthetic data and real datasets demonstrate improvements in stability and convergence speed of the GAN training, as well as higher sample quality. The approach also leads to higher classification accuracies in semi-supervised learning.

## 1 Introduction

Generative Adversarial Network (GAN) (Goodfellow et al., 2014) has been a new promising approach to unsupervised learning of complex high dimensional data in the last two years, with successful applications on image data (Isola et al., 2016; Shrivastava et al., 2016), and high potential for predictive representation learning (Mathieu et al., 2015) as well as reinforcement learning (Finn et al., 2016; Henderson et al., 2017). In a nutshell, GANs learn from unlabeled data by engaging the generative model ($G$) in an adversarial game with a discriminator ($D$). $D$ learns to tell apart fake data generated by $G$ from real data, while $G$ learns to fool $D$, having access to $D$'s input gradient.

Despite its success in generating high-quality data, such adversarial game setting also raises challenges for the training of GANs. Many architectures and techniques have been proposed (Radford et al., 2015; Salimans et al., 2016; Gulrajani et al., 2017) to reduce extreme failures and improve the sample quality of generated data. However, many theoretical and practical open problems still remain, which have impeded the ease-of-use of GANs in new problems. In particular, $G$ often fails to capture certain variation or modes in the real data distribution, while $D$ fails to exploit this failure to provide better training signal for $G$, leading to subtle mode collapse. Recently Arora et al. (2017) showed that the capacity of $D$ plays an essential role in giving $G$ sufficient learning guidances to model the complex real data distribution. With insufficient capacity, $D$ could fail to distinguish real and generated data distributions even when their Jensen-Shannon divergence or Wasserstein distance is not small.

In this work, we demonstrate that even with sufficient maximum capacity, $D$ might not allocate its capacity in a desirable way that facilitates convergence to a good equilibrium. We then propose a novel regularizer to guide $D$ to have a better model capacity allocation. Our regularizer is constructed to encourage $D$'s hidden binary activation patterns to have high joint entropy, based on a connection between the model capacity of a rectifier net and its internal binary activation patterns. Our experiments show that such high entropy representation leads to faster convergences, improved sample quality, as well as lower errors in semi-supervised learning.

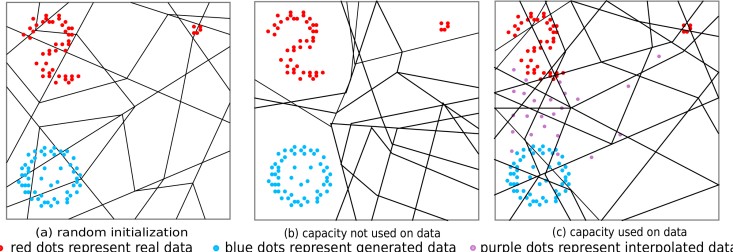

Figure 1: Capacity usage of the rectifier discriminator $D$ in different scenarios. $D$ (a rectifier net) cuts the input space into different linear regions, since rectifier nets compute piece-wise linear functions. **left**: $D$ uniformly spreads its capacity in the input space, but does not have enough capacity to distinguish all subtle variations within a data distribution. **middle**: $D$ uses its capacity in the region with no data; while real and fake data are correctly separated, variations within real data distribution are not represented by $D$, so cannot possibly be communicated to $G$ if this degeneracy persists through training; meanwhile all fake points in the same linear region passes the same gradient information to $G$, even if they are visually distinct. **right**: $D$ spends most capacity on real and fake data, but also in regions where $G$ might move its mass to in future iterations.

## 2 CAPACITY USAGE OF RECTIFIER NETS AND ITS EFFECTS ON GAN TRAINING

The motivation of our regularizer starts with an observation that during GAN training, the generator $G$ receives information about the input space *only* indirectly through the gradient of $D$, $\nabla_x D(x)$. Typically $D$ is a rectifier net. Absent of the last sigmoid nonlinearity, $D$ computes piecewise linear functions, meaning that the learning signal to $G$ is (almost) piecewise constant. The final sigmoid nonlinearity does not change the direction of input gradient in each linear region, but merely the scale of gradient vectors. The learning of $G$ in GANs can be interpreted as the movement of the fake samples generated by $G$ toward the real data distribution, guided by the (almost) piecewise constant vectorial signals according to input space partitioning by $D$. Hence, the diversity and informativeness of learning signals to $G$ is closely related to how the input space is partitioned, i.e. how $D$'s model capacity is allocated. In a region of the input space, how much capacity $D$ allocates into it can be approximately measured by the number of linear pieces in that region (Montufar et al., 2014). When $D$ spreads out its model capacity in a right way, the evenly dispersed partitioning helps $G$ to explore better and discover the real data manifold, while avoiding large unstable jumps due to overconfident extrapolation made by $D$.

Ideally, when GAN training is stable, the min-max game eventually forces $D$ to represent subtle variations in the real data distribution and pass this information for the learning of $G$. However, the discriminator $D$ is solely tasked to separate real samples from the generated fake ones. Thus $D$ has no incentive to do so, especially when the classification task for $D$ is too simple. Such is always the case in the early stage of the training and may persist to the later stage if the input space has high dimensionality or if $G$ already collapsed. In these situations, $D$ could overfit, and its internal layers could have degenerate representation whereby large portions of the input space are modelled as linear regions, as pictorially depicted in Fig. 1, and shown in the synthetic experiment in Sec. 4.1. With such degeneracy, learning signals from $D$ are not diverse and fail to capture the differences among different modes or subtle variations of the real data. Furthermore, such degeneracy could also cause the learning of $G$ to bluntly extrapolate, resulting in large updates, which in turn drops already discovered real data modes and/or leads to oscillations. We observe this phenomenon in the synthetic data problem in Sec. 4.1.

In this paper, we propose a new regularizer for training GANs where $D$ is a rectifier net. Our regularizer encourages the discriminator $D$ to cut the input space more finely around where the current $G$ distribution is supported, as well as where training might transport the generated data distribution to in the short future, as depicted in Fig. 1 (right). In this way, $G$ receives rich guidance for *faster exploration* and *more stable convergence*. The regularizer facilitates exploration because $D$ tells apart the generated fake samples from the real ones in *distinct* ways. This is because if the fake data points $x$ lie in different regions, learning signals $\nabla_x D(x)$ to $G$ are likely to point to different directions. Hence a concentrated mass in the fake data distribution has a better chance of been spread apart. On

the other hand, the convergence to equilibrium is more stable because there are less large piecewise linear regions, where $G$ learning constantly receives the same transportation direction, potentially leading to overshoot and oscillation. Our regularizer is constructed to encourage the activation patterns of the internal representations of points in a mini-batch to be diverse. As shown in Raghu et al. (2016), the different local linear regions defined by $D$ is closely related to the different activation patterns of $D$. In particular, two input points into $D$ with different activation patterns on all layers of $D$ are guaranteed to lie on different linear regions. More details are presented in Sec. 3 where the regularizer is defined, along with the analysis of its properties.

## 2.1 RELATED WORKS

Other regularization/gradient penalty techniques have also been proposed to stabilize GANs training (Gulrajani et al., 2017; Nagarajan & Kolter, 2017) recently. Gulrajani et al. (2017) adds an input gradient penalty to the update of $D$, so that the magnitude of signals passed to $G$ is controlled. Nagarajan & Kolter (2017) modifies the update of $G$ to avoid going where the magnitude of $\nabla_x D(x)$ is large. These methods, as well as other similar works that constrain the input gradient norm or the Lipschitz constant of $D$, all try to stabilize the training dynamics by regularizing the learning signal magnitude. This is different from our method that diversifies the learning signal directions. As discussed in the previous section, the diversified signal directions help both the convergence speed and the stability of the training. In Sec. 4.2, we empirically demonstrate that our proposed method achieves better results than Wasserstein GAN with gradient penalty (WGAN-GP) (Gulrajani et al., 2017).

The role of model capacity of the discriminator $D$ in training generative adversarial networks (GANs) has been previously explored by Arora et al. (2017). They show that $D$ with a finite number of parameters has limited capacity in distinguishing real data from the generated ones. They suggest increasing the discriminator $D$'s capacity among other modifications. Our work can be viewed as a continuation along this direction, except that we treat the model capacity not as a static number, but a dynamic function in the different parts of the input space during training. Because even with a large number of parameters, $D$ might not use its capacity in a right way to help convergence, as discussed previously. We explore the question on *where* and *how* $D$ can utilize its limited capacity effectively for better training convergence.

Encouraging $D$ to use its capacity in a constructive way is non-trivial. One theoretically sound potential approach to regularize $D$ is to use a Bayesian neural net, whose model capacity away from data is not degenerate. However, computationally scalable deep Bayesian neural networks are still an active area of research (Hernández-Lobato & Adams, 2015; Hasenclever et al., 2017) and are not easy to use. Alternatively, we can use auxiliary tasks to regularize $D$'s capacity usage. If given labelled data, semi-supervised learning as an auxiliary task for $D$, as shown in Salimans et al. (2016), improves GAN training stability and the resulting generative model. We hypothesize that if the data domain has other structures that can be exploited as supervised learning signal, Exploiting these structures could as well potentially improve the GAN training stability like in Salimans et al. (2016).

When no supervised task is available, auto-encoding is another potential possibility. Energy-Based GAN (EBGAN) (Zhao et al., 2016) and Boundary Equilibrium GAN (BEGAN) (Berthelot et al., 2017) use auto-encoders as their discriminators. However, both EBGAN and BEGAN have different objectives from the vanilla GAN. Furthermore, instead of using auto-encoding to simply regularize $D$, the auto-encoder loss is used to discriminate real data from fake ones. Hence, it is unclear if their benefits stem from the regularization effects or the alternative classification approach. Another set of works that use auto-encoding as an auxiliary task is in learning an inference network along with GAN (Donahue et al., 2016; Dumoulin et al., 2016). However, they both modify the input to $D$, so that $D$ classify not just the data, but together with the corresponding latent codes from $G$. Again, in this case, it is unclear if the regularization effect on the model capacity of $D$ is the source of any improvement in learning stability. Our preliminary results on using the auxiliary auto-encoding loss on real data show that it does not lead to improvement (see Discussion and Future Work in Sec. 5).

## 3   BINARIZED REPRESENTATION ENTROPY

We now introduce our regularizer, the binarized representation entropy regularizer (BRE). Recall that we would like to encourage diverse activation patterns in $D$ across different samples. For a sample $x$, its activation pattern on a particular internal layer of $D$ can be represented by a binary vector, as shown in Figure 2. In particular, let $\mathbf{h} \in \mathbb{R}^d$ be the immediate pre-nonlinearity activity of a sample $x$ on a particular layer of $d$ hidden units [1]. The activation pattern of $x$ on this layer can be represented by the sign vector of $\mathbf{h}$, defined as $\mathbf{s} = \text{sign}(\mathbf{h}) := \frac{\mathbf{h}}{|\mathbf{h}|} \in \{\pm 1\}^d$ where $|\cdot|$ is entry-wise absolute value.

We call this binary vector $\mathbf{s} \in \{\pm 1\}^d$ the activation vector of the sample $x$ on this particular layer.

In this work, we model the activation vector of each sample in a particular layer of $D$ as a random binary vector. Given a mini-batch, $\{x_1, \ldots, x_K\}$ of size $K$, assume that each binary activation vector $\mathbf{s}_k$ of $x_k$, $k = 1, \ldots, K$, on a particular layer with $d$ hidden units is an independent sample of a random binary vector $U = (U_1, \ldots, U_d)$, where $U_i$ denotes a Bernoulli random variable[2] with parameters $p_i$ and distribution function $\mathbb{P}_i$ for $i = 1, \ldots, d$. Also denotes the joint distribution function of

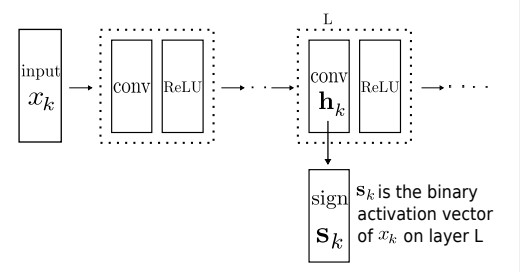

Figure 2: Activation vector $\mathbf{s}_k$ of a sample $x_k$ on a layer $L$ immediately before nonlinearity.

$(U_1, \ldots, U_d)$ by $\mathbb{P}$. To have diverse activation patterns, we would like to construct a regularizer that encourages $\mathbb{P}$ to have a large joint entropy. Ideally, one could use an empirical estimate of the entropy function as a desired regularizer. However, sample-based estimation of the entropy of a high-dimensional distribution has been well known to be difficult, especially with a small mini-batch size (Darbellay & Vajda, 1999; Miller, 2003; Kybic, 2007; Kybic & Vnučko, 2012; Scott, 2015).

We instead propose a simple *binarized representation entropy* (BRE) regularizer, which encourages the entropy of $\mathbb{P}$ to be larger (in a weak manner). For a particular layer in $D$, our BRE regularizer $R_{\text{BRE}}$ is computed over a mini-batch of $\{x_1, \ldots, x_K\}$, and consists of two terms, *marginal entropy* $R_{\text{ME}}$ and *activation correlation* $R_{\text{AC}}$, both acting on the binarized activation vectors of the hidden units[3]: $R_{\text{BRE}} = R_{\text{ME}} + R_{\text{AC}}$, where

$$R_{\text{ME}} = \frac{1}{d} \sum_{i=1}^{d} \bar{\mathbf{s}}_{(i)}^2 = \frac{1}{d} \sum_{i=1}^{d} \left( \frac{1}{K} \sum_{k=1}^{K} \mathbf{s}_{k,i} \right)^2 ; \quad \text{and} \quad R_{\text{AC}} = \frac{1}{K(K-1)} \sum_{\substack{j,k=1 \\ j \neq k}}^{K} \frac{|\mathbf{s}_j^\top \mathbf{s}_k|}{d}. \quad (1)$$

Here $\bar{\mathbf{s}}_{(i)} = \frac{1}{K} \sum_{k=1}^{K} \mathbf{s}_{k,i}$ is the average of the $i$th element (corresponding to the $i$th hidden unit) of the activation vectors $\mathbf{s}_k$ across the mini-batch, where $\mathbf{s}_k$ is the activation vector of $x_k$ for $k = 1, \ldots, K$, as shown in Figure 3. Thus $R_{\text{ME}}$ can be interpreted as an empirical estimate of $\frac{1}{d} \sum_{i=1}^{d} \mathbb{E}[U_i]^2$, and $R_{\text{AC}}$ as an empirical estimate of $\frac{1}{d} \mathbb{E}[|U^\top \tilde{U}|] = \frac{1}{d} \mathbb{E}[|\sum_{i=1}^{d} U_i \tilde{U}_i|]$, where $U$, $\tilde{U}$ are two i.i.d. random vectors with probability function $\mathbb{P}$.

As shown in Section 3.1, our regularizer encourages a large joint entropy of this random binary vector. In particular, we show that the first term, $R_{\text{ME}}$, encourages individual hidden units to be active half of the time on average, to have high *marginal entropy*; the second term, $R_{\text{AC}}$, encourages low *activation correlation* between each pair

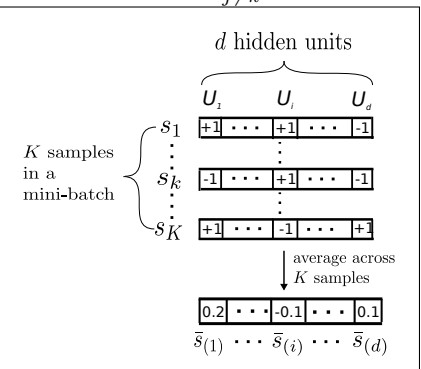

Figure 3: Notations for defining $R_{\text{BRE}}$.

---

[1]We use column vectors in this paper.

[2]The Bernoulli distribution is defined over $\{+1, -1\}$ instead of $\{0, 1\}$.

[3]We may apply this regularizer to multiple layers in $D$. In that case, we will sum all the $R_{\text{BRE}}$'s of each layer.

of the hidden units. We further show in Section 3.2 that having the regularizer being close to 0 is a necessary condition for $(U_1, \ldots, U_d)$ to achieve its maximum entropy. Details in the practical implementation of our regularizer are discussed in Section 3.3.

## 3.1 BRE ENCOURAGES HIGH JOINT ENTROPY OF THE ACTIVATION PATTERNS

Note that each summand $\bar{\mathbf{s}}_{(i)}$ in $R_{\mathrm{ME}}$ is an empirical estimate of $2p_i - 1$, the mean of the marginal distribution $\mathbb{P}_i$. Thus minimizing $\bar{\mathbf{s}}_{(i)}^2$ leads to $p_i = \frac{1}{2}$, i.e. $U_i$ is zero-mean for $i = 1, \ldots, d$. In other words, $R_{\mathrm{ME}}$ is 0 when there are equal number of $\pm 1$ in $U$.

Moreover, for $j, k = 1, \ldots, K$ where $j \neq k$, minimizing $|\mathbf{s}_j^\top \mathbf{s}_k|$ in the second term $R_{\mathrm{AC}}$ is essentially equivalent to minimizing $\left(\mathbf{s}_j^\top \mathbf{s}_k\right)^2$. Thus, minimizing $R_{\mathrm{AC}}$ can be seen as minimizing $\mathbb{E}\left[\left(U^\top \tilde{U}\right)^2\right]$ where $U, \tilde{U}$ are i.i.d. from $\mathbb{P}$. Since minimizing $R_{\mathrm{ME}}$ is enforcing $U_i$ to be zero-mean for $i = 1, \ldots, d$, as shown in Proposition 3.1, minimizing $R_{\mathrm{AC}}$ enforces the pairwise independence of the $U_i$'s.

Lastly, Assuming the hidden units $U_i$'s are zero-mean and pairwise independent, by Corollary 3.3 of Gavinsky & Pudlák (2015) (which we restate in Appendix C for completeness), we have that the entropy of $\mathbb{P}$ satisfies

$$H(\mathbb{P}) \geq \log(d + 1).$$

**Proposition 3.1.** *Let* $U = (U_1, \ldots, U_d)$ *be a zero-mean multivariate Bernoulli vector of* $\mathbb{P}$, *and* $\tilde{U} = (\tilde{U}_1, \ldots, \tilde{U}_d)$ *denotes another random vector of* $\mathbb{P}$ *that is independent to* $U$. *Then*

$$\mathbb{E}\left[\left(U^\top \tilde{U}\right)^2\right] = \mathbb{E}\left[\left(\sum_{i=1}^d U_i \tilde{U}_i\right)^2\right] = d + \sum_{\substack{i,t=1 \\ i \neq t}}^d Cov\left(U_i, U_t\right)^2.$$

We defer the proof of this proposition to Appendix B.

## 3.2 MAXIMUM ENTROPY REPRESENTATION HAS $R_{\mathrm{BRE}} \approx 0$.

We further show that $R_{\mathrm{BRE}} \approx 0$ is a necessary condition for $\mathbb{P}$ to achieve the maximum entropy.

It is straightforward that the maximum entropy of $\mathbb{P}$ is achieved if and only if $\mathbb{E}\left[U_i\right] = 0$ $(p_i = 1/2)$ for all $i \in \{1, ..., d\}$, i.e. each hidden unit is activated half of the time, and $(U_1, \ldots, U_d)$ are mutually independent. Therefore, the $i$th element of the average activation vector $\bar{\mathbf{s}}_{(i)}$ is approximately zero for $i \in \{1, ..., d\}$, and so is $R_{\mathrm{ME}}$.

Further, note that $R_{\mathrm{AC}}$ is an empirical estimate of $\mathbb{E}\left[\left|\sum_{i=1}^d M_i/d\right|\right]$ where $M_i = U_i \tilde{U}_i$ for $i = 1, \ldots, d$. Note that given $p_i = 1/2$ and $U_i$'s are mutually independent, one can show that $M_i$'s are mutually independent and have the distribution of Bernoulli(0.5) as well. Therefore by the Central Limit Theorem, the distribution of $\sum_{i=1}^d M_i$ converges in distribution to the Gaussian distribution $\mathcal{N}(0, d)$. Given sufficiently large $d$, the distribution of $\frac{\sum_{i=1}^d M_i}{d}$ is approximately $\mathcal{N}(0, 1/d)$, and thus $R_{\mathrm{AC}}$ is approximately zero[4].

## 3.3 GAN TRAINING WITH BRE REGULARIZER

### PRACTICAL IMPLEMENTATIONS OF $R_{BRE}$

In practice, due to the degenerate gradient of the sign function, we replace $\mathbf{s}$ in $R_{\mathrm{ME}}$ by its smooth approximation $\mathbf{a} = \operatorname{softsign}(\mathbf{h}) := \frac{\mathbf{h}}{|\mathbf{h}| + \epsilon}$, where $\epsilon$ is a hyperparameter to be chosen. If $\epsilon$ is too small, the nonlinearity becomes too non-smooth for stochastic gradient descent; if it is too large, it fails to be a good approximation to the sign function. Furthermore, not only different layers could have different scales of $\mathbf{h}$, hence requiring different $\epsilon$, during training the scale of $\mathbf{h}$ could change

---

[4]Note that the expectation of $R_{AC}$ under the maximum entropy assumption is not zero, but a small number on the order of $1e - 3$.

too. Therefore, instead of setting a fixed $\epsilon$, we set $\epsilon = \zeta \, \text{avg}(|\mathbf{h}|)$, where $\zeta$ is some small constant and $\text{avg}(|\mathbf{h}|)$ is a scalar, where the average runs over samples in the minibatch and the $d$ dimensions of the layer. In this way, softsign$(\cdot)$ is invariant with respect to any multiplicative scaling of $\mathbf{h}$ in the forward pass of the computation; in the backward pass for the gradient computation, we do not backpropagate through $\epsilon$. We choose $\zeta = 0.001$, as we observe empirically that this usually makes $90\%$ to $99\%$ of units to have absolute value at least .9. An alternative to this softsign is the tanh nonlinearity. However, tanh lacks the scale invariance of our proposed softsign with varying $\epsilon$, hence potentially less effective in capturing the nature of input space partitioning. In Sec. 4.2 (Table. 1), we confirm empirically that using tanh instead of softsign decreases the effectiveness of BRE.

We also relax $R_{AC}$ by allowing a soft margin term, as $R_{AC} = \text{avg}_{j \neq k} \max \left(0, |\mathbf{a}_j^\top \mathbf{a}_k|/d - \eta\right)$. Recall that $\mathbf{a}_j^\top \mathbf{a}_k/d$ has an approximate distribution of $N(0, 1/d)$, so a good choice for the margin threshold is $\eta = c\sqrt{1/d}$, where we adopt the "$3\sigma$ rule" and choose $c = 3$ to leave $99.7\%$ of $i, j$ pairs unpenalized in the maximum entropy case.

To regularize GAN training, $R_{BRE}$ is applied to the immediate pre-nonlinearity activities on selected layers of $D$. Therefore, if there is any normalization layer before nonlinearity, $R_{BRE}$ needs to be applied after the normalization. We emphasize that we use softsign for the regularizer only, we do not modify the nonlinearity or any other structure of the neural net.

### WHICH LAYERS SHOULD $R_{BRE}$ BE APPLIED ON?

Technically $R_{BRE}$ can be applied on any rectifier layer before the nonlinearity. However, having it on the last hidden rectifier layer before classification might hinder $D$'s ability to separate real from fake, as the high entropy representation encouraged by $R_{BRE}$ might not be compatible with linear classification. Therefore, for unsupervised learning, we apply $R_{BRE}$ on all except the last rectifier nonlinearity before the final classification; for semi-supervised tasks using the augmented class setup from Salimans et al. (2016), we apply $R_{BRE}$ only on 2nd, 4th and 6th convolutional layer, and leave the three nonlinear layers before the final softmax untouched.

### WHICH PART OF THE DATA SHOULD $R_{BRE}$ BE APPLIED ON?

Recall from Sec. 2 that we want $D$ to spend enough capacity on both the real data manifold, and the current generated data manifold by $G$, as well as having adequate capacity in region where we do not currently observe real or fake points but might in future iterations. To enforce this, we apply $R_{BRE}$ on generated data minibatch, as well as random interpolation inbetween real and generated data. Specifically, let $\mathbf{x}_k$ and $\tilde{\mathbf{x}}_k$ be a real and a fake data points respectively, we sample $\alpha_k \sim U(0, 1)$ and let $\hat{\mathbf{x}}_k = \alpha_k \mathbf{x}_k + (1 - \alpha_k)\tilde{\mathbf{x}}_k$, and apply $R_{BRE}$ on selected layer representation computed on interpolated data points $\{\hat{\mathbf{x}}_k \mid k = 1 \ldots, K\}$ as well.

## 4 EXPERIMENTS

Using a 2D synthetic dataset and CIFAR10 dataset (Krizhevsky, 2009), we show that our BRE improves unsupervised GAN training in two ways: (a) when GAN training is unstable (for e.g. due to architectures that are less well tuned than DCGAN (Radford et al., 2015)), BRE stabilizes the training and achieves much-improved results, often surpassing tuned configurations. (b) with architecture and hyperparameters settings that are engineered to be stable already, BRE makes GAN learning converges faster. We then demonstrate that BRE regularization improves semi-supervised classification accuracy on CIFAR10 and SVHN dataset (Netzer et al., 2011). Additional results on imbalanced 2D mixture as well as CelebA dataset are presented in the Appendix D.

### 4.1 SYNTHETIC DATASET: MIXTURE OF GAUSSIANS

We first demonstrate BRE regularizer's effect on fitting a 2D mixture of Gaussian. In Figure 4, the top three rows and bottom three ones correspond respectively to experiments without the BRE regularizer (control) and with the regularizer (treat). Within each setting, each row represents one iteration during GAN training, selected to be at the beginning, middle, and the end of the training process. The first column shows real data points (blue) and generated data points (red). The second to fifth columns show hidden layers 1 to 4 of $D$, where contiguous pixels with the same colour have the same binary activation pattern on that particular layer. The last column shows the probability

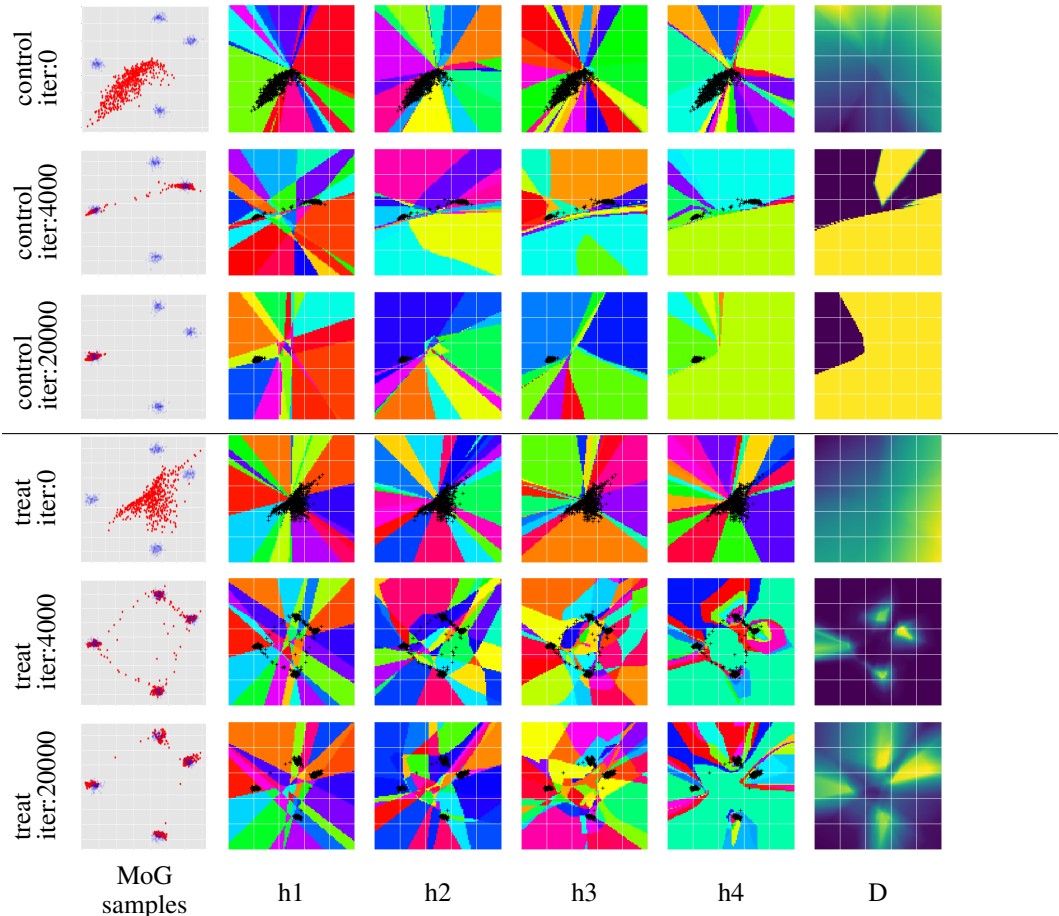

Figure 4: Fitting 2D Mixture of Gaussian (MoG): $h1$-$h4$ show the input space linear region defined by different binary activation patterns on each layer; each colour corresponds to one unique binary pattern; the last column shows probability of being real according to $D$ ; BRE is applied on $h2$ and $h3$. Experimental details and more visualization in Appendix A and D, including one set of comparison for fitting an imbalanced mixture in Fig. 16-17.

of real data according to $D$ . The BRE regularizer is added on layers $h2$ and $h3$. More results in Appendix D.

By adding BRE, the input domain is partitioned more finely as reflected by visualization for layers $h2$, $h3$ and $h4$. The richer $D$ representation allows more effective exploration of different input regions because the gradient signals provided by $D$ to $G$ are more diverse than the degenerate baseline case where $D$ is linear in large regions of the input. Once a real data mode is discovered, $G$ locks onto it without oscillation. This shows that with better $D$ capacity usage, the GAN optimisation converges faster and is more stable, while the resulting equilibrium suffers much less from mode dropping.

## 4.2 FASTER AND BETTER CONVERGENCE IN UNSUPERVISED LEARNING

We quantitatively measure the resulting $G$ using the Inception score (Salimans et al., 2016).

Table 1 shows improved final Inception scores on DCGAN, as well as the following non-standard architectures (only mentioning difference from the standard DCGAN): densely connected convnet Huang et al. (2016) for $D$ ; $G$ and $D$ with an equal number of filters on each layer; $D$ with ReLU nonlinearity. In all cases, the models regularized by BRE improve over the baseline counter-parts without regularization (no-BRE). Furthermore, vanilla GAN's with BRE applied on multiple $D$ layers (BRE_multi) always outperform WGAN-GP (Gulrajani et al., 2017). Fig. 6 shows some

generated samples from a DCGAN-ReLU model without and with BRE regularization. Fig. 9 of Appendix D.1 show more samples laid out by t-SNE visualization to better illustrate mode collapsing.

| | densenet $D$ | | ReLU $D$ |
|---|---|---|---|
| WGAN-GP BRE_single | $3.9589 \pm 0.6632$ | ln WGAN-GP no-BRE | $4.4359 \pm 0.2975$ |
| WGAN-GP no-BRE | $4.1046 \pm 0.3443$ | no-BRE | $5.5409 \pm 0.2363$ |
| no-BRE | $6.3662 \pm 0.1465$ | WGAN-GP no-BRE | $5.9606 \pm 0.3584$ |
| BRE_multi | $6.5650 \pm 0.1979$ | WGAN-GP BRE_single | $6.2105 \pm 0.3607$ |
| **BRE_single** | **$6.6261 \pm 0.1529$** | BRE_single | $6.2526 \pm 0.2239$ |
| | Equal Size $G$ and $D$ | BRE_multi_tanh | $6.3754 \pm 0.2870$ |
| | | **BRE_multi** | **$6.7715 \pm 0.3162$** |
| no-BRE | $5.1330 \pm 0.5491$ | | |
| BRE_single | $6.0375 \pm 0.3669$ | | DCGAN |
| BRE_multi_tanh | $6.3455 \pm 0.2132$ | WGAN-GP no-BRE | $6.3284 \pm 0.4642$ |
| WGAN-GP BRE_single | $6.4515 \pm 0.2315$ | no-BRE | $6.5865 \pm 0.1837$ |
| WGAN-GP no-BRE | $6.6993 \pm 0.1705$ | BRE_single | $6.6908 \pm 0.2539$ |
| **BRE_multi** | **$7.0569 \pm 0.2031$** | **BRE_multi** | **$6.7312 \pm 0.1365$** |

Table 1: BRE on various architectures: no-BRE is the baseline in each case; with BRE weight in other cases is set to $1$.; *single* and *multi* signify whether BRE is applied on one layer in the middle of $D$ or multiple (see Appendix A for more details); *ln* for layer normalization in $G$ and $D$ (default is batchnorm); *tanh* means the softsign nonlinearity in BRE is replaced by tanh.

Fig. 5 shows that with BRE, DCGAN training converges faster, as measured by Inception score. The $1\sigma$ error bars are estimated from ten different random runs. Because DCGAN architecture is engineered to be stable, in the end, baseline DCGAN can still achieve comparable Inception score with the regularized version on average. But clearly the convergence is much faster with BRE during the initial transient phase, confirming our intuition that BRE improves exploration. Fig. 7 shows Inception score and (thresholded) activation correlation values ($R_{\text{AC}}$) during one particular set of runs with default DCGAN optimization settings, and a more aggressive optimization setting. In both cases, BRE regularization results in similarly faster convergence, and higher final Inception score in the unstable case with more aggressive optimization. The bottom row in Fig. 7 shows that BRE regularization is indeed making a qualitative difference to the activation correlation ($R_{\text{AC}}$) by keeping it low during training in both cases.

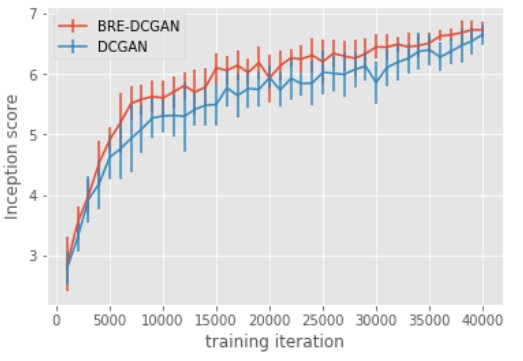
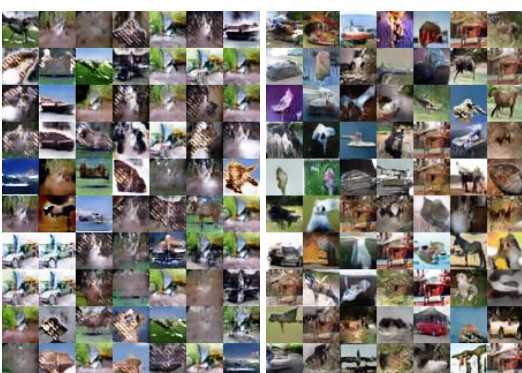

Figure 5: Even with stable DCGAN architecture, BRE makes convergence faster.

Figure 6: Samples for DCGAN-ReLU without BRE (left) vs with BRE (right)

### 4.3 IMPROVED SEMI-SUPERVISED LEARNING ON CIFAR10 AND SVHN

BRE regularization is not only compatible with semi-supervised learning using GAN's, but also improves classification accuracy.

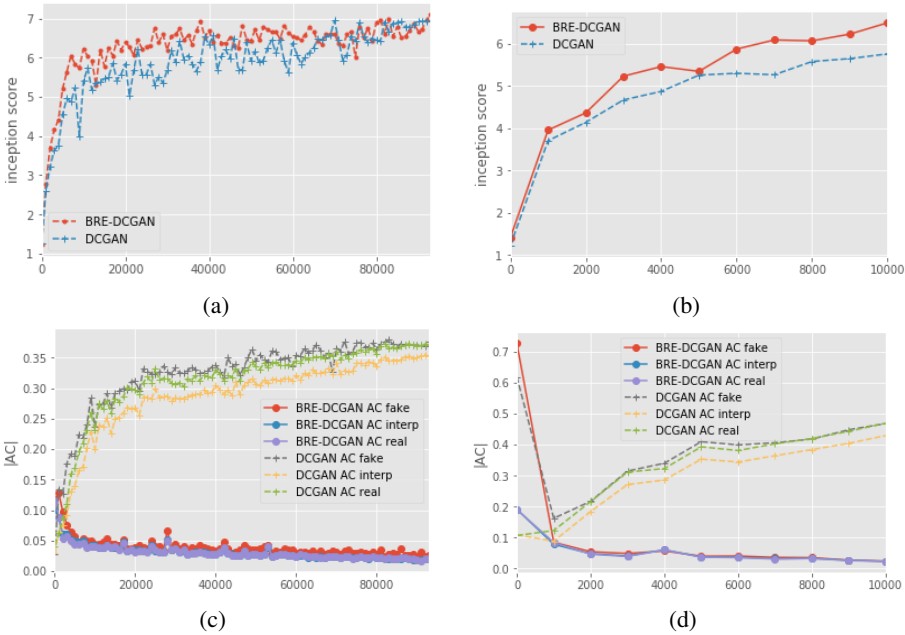

Figure 7: Inception scores and regularizer values during training: (left column, i.e. (a) and (c)) default optimization setting; (right column, i.e. (b) and (d)) more aggressive optimization. Details in Appendix A. (top row, i.e. (a) and (b)) Inception scores during training; (bottom row, i.e. (c) and (d)) $R_{AC}$ term of BRE on fake, real, and interpolation inbetween. Even though BRE is not applied on real, model still allocates enough capacity when BRE is applied on fake and interpolation.

Table. 2 shows results on CIFAR10 with feature matching semi-supervised learning GAN. BRE allows the learning process to discover a better solution during training that also generalizes better, indicated by a lower training classification loss as well as lower test classification error rates. We used the same code and hyperparameters [5] from Salimans et al. (2016). Details on BRE hyperparameters are in Appendix A, learning curve plots in Appendix D.2.

On Street View House Numbers (SVHN) dataset (Netzer et al., 2011), with the same setup from Salimans et al. (2016), learning is not always stable when trained for a long time. Fig. 8 (top row) shows that without BRE regularization, when trained for a very long time, sometimes learning diverges. Such failure is dramatically reduced by BRE (bottom row of Fig. 8).

|  | Test error rate (%) | Train classification loss |
|---|---|---|
| FM (reported in Salimans et al. (2016)) | $18.63 \pm 2.32$ | |
| FM, 10 ensemble (reported in Salimans et al. (2016)) | $15.59 \pm 0.47$ | |
| FM (our run) | $17.42 \pm 0.50$ | $9.25\mathrm{e}{-4} \pm 5.05\mathrm{e}{-4}$ |
| FM + BRE | $16.98 \pm 0.52$ | $5.03\mathrm{e}{-4} \pm 3.50\mathrm{e}{-4}$ |
| FM, 10 ensemble (our run) | $14.25$ | |
| FM + BRE, 10 ensemble | $13.93$ | |

Table 2: Semi supervised learning on CIFAR10: feature matching (FM) from Salimans et al. (2016)); 1000 labeled training examples.

## 5 DISCUSSION AND FUTURE WORK

There are still many unexplored avenues along this line of research. For example, how can our new regularizer collaborate with other GANs training techniques to further improve the training GANs?

---

[5]https://github.com/openai/improved-gan

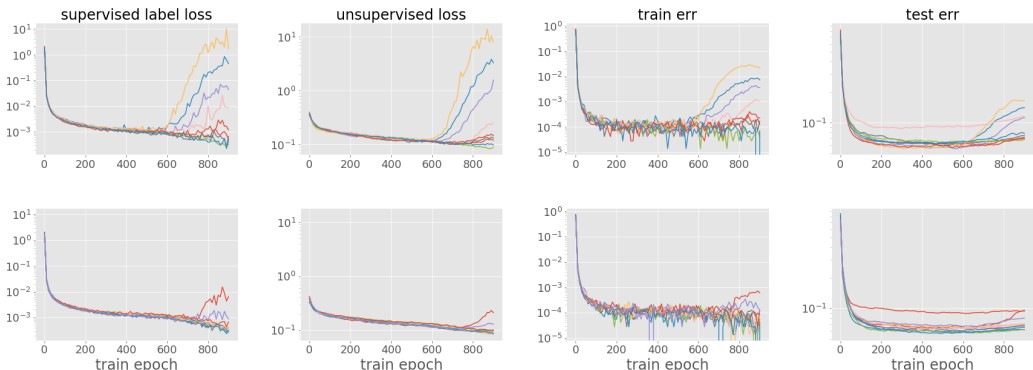

Figure 8: Improved semi-supervised learning on SVHN: each curve corresponds to a different random seeding; we repeat the same set of seeds for runs without BRE (Top row), and with BRE (Bottom column); both the random seed for selecting labeled examples and random seeds for model parameter initialization are varied.

We leave such further explorations for future works. Meanwhile, there are two interesting questions related to the central theme of this work.

DOES DIRECTLY REGULARIZING THE DIVERSITY OF $\nabla_x D(x)$ WORK? IF NOT, WHY NOT?

To diversify $G$ 's learning signals, it might be tempting to enforce gradient directions $\nabla_x D(x_k)$ to be diverse. However, in rectifier networks, if two inputs share the same activation pattern, the input gradients located at the two points are co-linear; hence any gradient-based learning with such diversity regularizer would have difficulty pulling them apart. In general, unlike BRE which operates directly on both activated and non-activated portions of $D$ 's internal units, an input gradient regularizer can only access information on the activated path in the network, so that it can only encourage existing non-shared activated path, but cannot directly create any new non-shared activated path. In theory, tanh nonlinearities as activations for $D$ could avoid this problem, but such network is hard to train in the first place. In our preliminary studies, on networks with tanh, input gradient diversity regularizer with either cosine similarity or a soft-sign based regularizer like BRE does not work.

COULD AUXILIARY TASKS HELP REGULARIZE $D$ ?

As discussed in Sec. 2.1, auxilary tasks could potentially regularize $D$ and stabilize training. One possible auxilary task is reconstruction loss. We performed some preliminary experiments, and found that reconstructing real data as auxilary tasks worsens the resulting learned $G$ . See Appendix. D.3 for results. Further study is needed, and is beyond the scope of this work.

## 6 CONCLUSIONS

We proposed a novel regularizer in this paper to guide the discriminator in GANs to better allocate its model capacity. Based on the relation between the model capacity and the activation pattern of the network, we constructed our regularizer to encourage a high joint entropy of the activation pattern on the hidden layers of the discriminator $D$. Experimental results demonstrated the benefits of our new regularizer: faster progress in the initial phase of learning thanks to improved exploration, more stable convergence, and better final results in both unsupervised and semi-supervised learning.

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

## A    Model and hyperparameter details

### A.1    2D example

$G$ is 4-layer (excluding noise layer) MLP with ReLU hidden activation function, and tanh visible activation; $D$ is 5-layer MLP with LeakyRelu(.2). Both $D$ and $G$ has 10 units on each hidden layer, no batch or other normalization is used in $D$ or $G$ , no other stabilization techniques are used; For Fig. 4, Fig. 12, and Fig. 13, lr=.001 with adam(.0, .999), and BRE regularizer weight 1., applied on h2 and h3; both lr and BRE weight linearly decay to over iterations to $1e-6$ and 0 respectively. For Fig. 12 and Fig. 13, lr=.002 with adam(.5, .999), and BRE regularizer weight 1., applied on h2.

### A.2    Unsupervised Learning CIFAR10

Table. 1, "single" means that BRE is applied on a single layer (the middle one of all nonlinear layers of $D$ ), while "multi" means all nonlinear layers except the first one and last two (the final classification and the nonlinear layer before it).

For Fig. 7, the default optimization setting (left column, i.e. (a) and (c)) is $lr = 2e{-}4$ and one $D$ update per $G$ update, lr for both $D$ and $G$ annealed to $1e-6$ over $90K$ $G$ updates; while the aggressive setting (right column, i.e. (b) and (d)) is $lr = 2e{-}3$ and three $D$ update for every $G$ update, lr for both $D$ and $G$ annealed to $1e-6$ over $10K$ $G$ updates.

### A.3    Semi-Supervised Learning

We used exactly the same code and GAN hyperparameters [6] from Salimans et al. (2016). $R_{BRE}$ regularization is applied on every other second layer, starting from the 2nd until 4 layers before the classification layer (applied on three layers in total). On CIFAR10, we used a regularizer weight of .01, and on SVHN we used 0.1. BRE is applied on real, fake and interp data.

## B    Proof of Proposition 3.1

*Proof.* Let $M_i = U_i \tilde{U}_i$. Then

$$\mathbb{E}\left[\left(\sum_{i=1}^d U_i \tilde{U}_i\right)^2\right] = \mathbb{E}\left[\left(\sum_{i=1}^d M_i\right)^2\right]$$

$$= \sum_{i=1}^d \mathbb{E}\left[M_i^2\right] + \sum_{i\neq t} \mathbb{E}\left[M_i M_t\right]$$

$$= \sum_{i=1}^d \mathbb{E}\left[U_i^2 \tilde{U}_i^2\right] + \sum_{i\neq t} \mathbb{E}\left[U_i \tilde{U}_i U_t \tilde{U}_t\right]$$

$$\overset{(1)}{=} \sum_{i=1}^d \mathbb{E}\left[U_i^2\right]^2 + \sum_{i\neq t} \mathbb{E}\left[U_i U_t\right]^2$$

$$\overset{(2)}{=} d + \sum_{i\neq t} \mathbb{E}\left[U_i U_t\right]^2$$

$$\overset{(3)}{=} d + \sum_{i\neq t} \mathrm{Cov}\left(U_i, U_t\right)^2,$$

where Equation (1) is due to the independence of $U$ and $\tilde{U}$, Equation (2) is due to that $U_i^2 = 1$ with probability 1, and Equation (3) is because $\mathbb{E}\left[U_i\right] = 0$.    $\square$

---

[6]https://github.com/openai/improved-gan

## C    COROLLARY 3.3 OF GAVINSKY & PUDLÁK (2015)

**Theorem   C.1**   (Corollary   3.3   of   Gavinsky   &   Pudlák   (2015)).   *Let*   $H_{\min}(\mathbb{P})$   $=$
$-\log(\max_x \mathbb{P}(X = x))$.   *Also let*   $(U_1, \ldots, U_d)$   *be pairwise independent random variable of*
*Bernoulli(*0.5*). Then,*

$$H(\mathbb{P}) \geq H_{\min}(\mathbb{P}) \geq \log(d + 1).$$

## D    ADDITIONAL EXPERIMENTAL RESULTS

### D.1    MORE SAMPLES FROM DCGAN-RELU

We show more samples generated from the DCGAN-ReLU model, mentioned in Sec. 4.2, without
the BRE regularizer (with red frames) and with the BRE regularizer (with blue frames) in Fig. 9.
In Fig. 9 images are arranged based on $L_2$ distances in the original pixel value space. Images that
are similar in pixel values are roughly grouped together. To achieve this, we apply t-SNE (Maaten
& Hinton, 2008) to reduce the dimensionality of these images into 2D points. These 2D coordinates
are then transformed to 2D grids RasterFairy [7]. At the same time, the neighborhood relations of the
rastered 2D points are preserved to a certain degree. We then use these rastered 2D points to arrange
the location of these images.

### D.2    SEMI-SUPERVISED LEARNING CURVES

Fig. 10 shows the Learning curves for semi-supervised learning on CIFAR10.

### D.3    RECONSTRUCTION AS AUXILIARY TASK TO REGULARIZE $D$ WORSENS RESULTS

|  | D Recon, no BRE |
| --- | --- |
| ln, $\lambda_{recon} = 10$ | $6.1958 \pm 0.2438$ |
| ln, $\lambda_{recon} = 1$ | $6.2218 \pm 0.2390$ |
| ln, $\lambda_{recon} = .1$ | $6.2437 \pm 0.2346$ |
| ln, $\lambda_{recon} = 0$ | $6.4025 \pm 0.2187$ |
| bn, $\lambda_{recon} = 1$ | $6.5356 \pm 0.2176$ |
| bn, $\lambda_{recon} = .1$ | $6.5475 \pm 0.2798$ |
| bn, $\lambda_{recon} = 0$ | $6.5865 \pm 0.1837$ |

Table 3: Reconstruction as an auxiliary task worsens results. $\lambda_{recon}$ is the weight of the $l_2$ recon-
struction loss term. With both batch or layer normalization, reconstruction auxiliary task hurts the
final results.

### D.4    CELEBA

We compare stable and unstable runs of DCGAN on CelebA dataset (Liu et al., 2015), as well as
the effect of the BRE regularizer. Fig. 11(a) shows thresholded $R_{AC}$ term (defined in Sec. 3.3)
through training. The model being investigated is a 4-layer DCGAN for both $G$ and $D$ , with batch
normalization. The unstable run (Fig. 11(b)) uses a large initial learning rate of .01 and 3 $D$ update
steps for each $G$ update, whereas the stable run (Fig. 11(d)) uses initial $lr = 2e - 3$ and 1 $D$ update
for each $G$ update. Even without the BRE regularizer, we can see that when GAN training is stable,
$D$ uses more capacity around fake and real data as well as inbetween, as measured by $R_{AC}$ values
in Fig. 11(a). When BRE regularizer is applied, the usage of $D$ 's capacity is more improved, and
resulted in more diversity in the learned distribution by $G$ (Fig. 11(c)).

### D.5    ADDITIONAL 2D MOG RESULTS

We show more 2D mixture of Gaussian results in Fig. 12, 13, 14, and 15.

---
[7]https://github.com/Quasimondo/RasterFairy

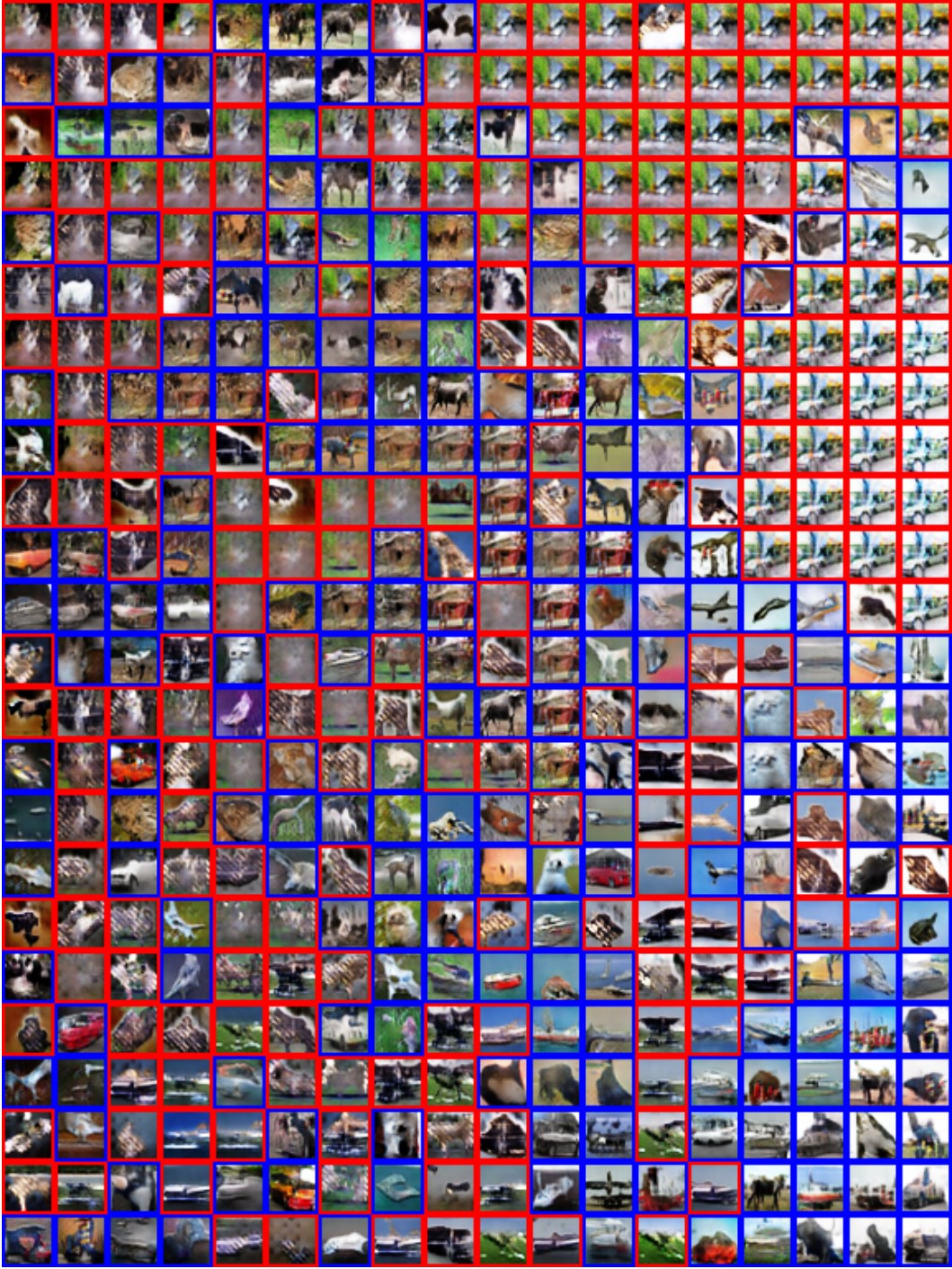

Figure 9: Rastered t-SNE visualization of DCGAN-ReLU CIFAR10 samples. Images with red frames are generated without BRE and images with blue frames are generated with BRE. Locations roughly indicates similarity between images in the pixel value space.

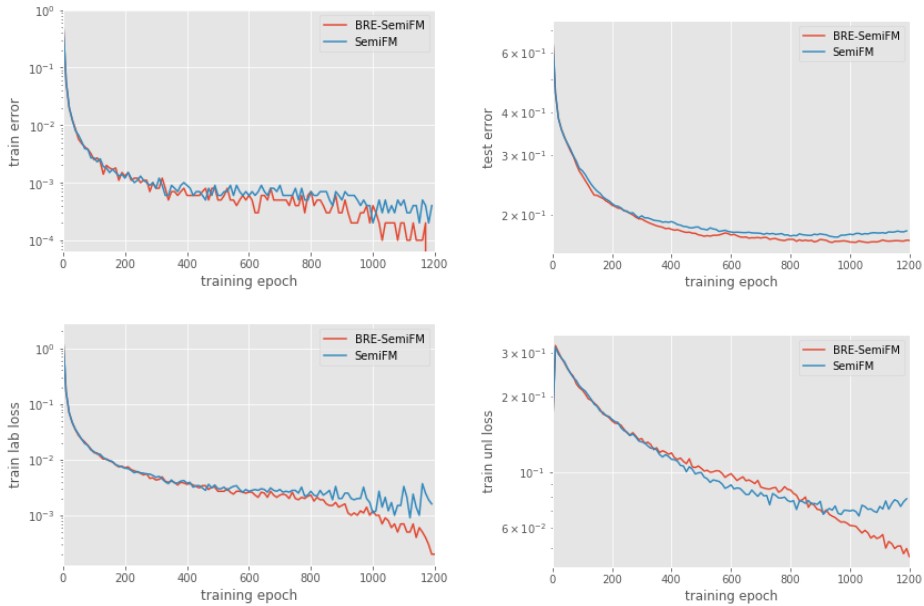

Figure 10: Improved semi-supervised learning on CIFAR-10: BRE regularizer placed on every other second layer, starting from the 2nd until 4 layers before the classification layer. Regularizer weight is .01 and not decayed.

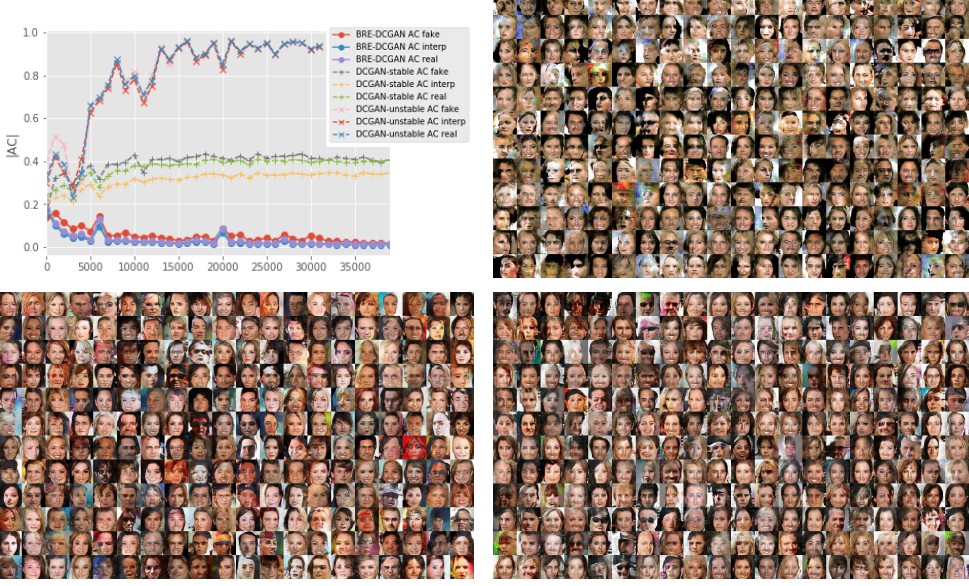

Figure 11: (Thresholded) Activity correlation (AC) values (top left) and samples at iteration 10K: (top right) DCGAN unstable run ($lr = .01$ and $3$ $D$ update steps for each $G$ update); (lower right) DCGAN stable run ($lr = 2e - 3$ and $1$ $D$ update steps for each $G$ update); (lower left) BRE-DCGAN, DCGAN training with BRE regularizer same hyperparameters as DCGAN stable run in lower right plot. BRE-DCGAN results are visibly more diverse.

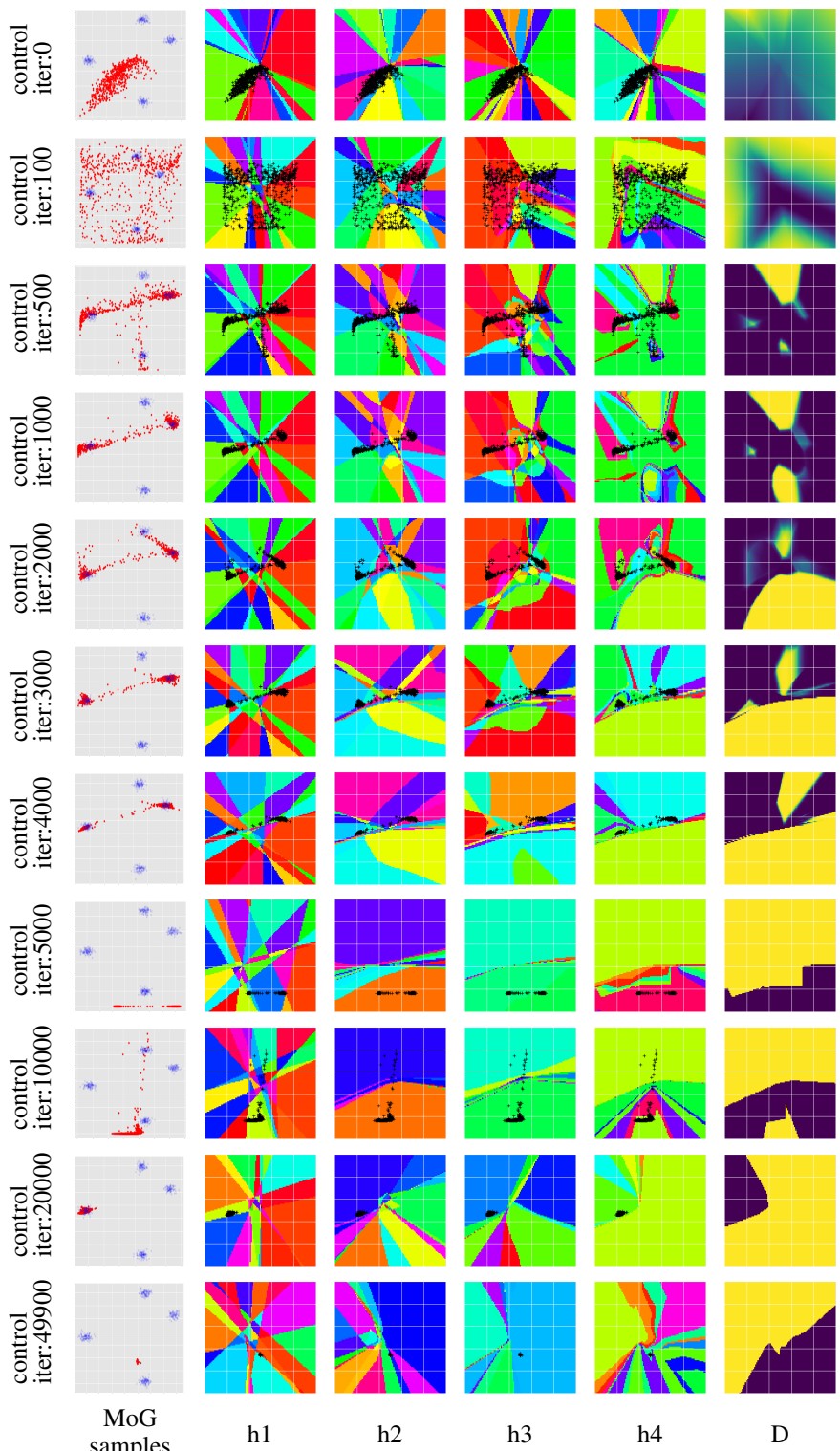

Figure 12: More Results on Fitting 2D Mixture of Gaussian on the control group. See Figure 4 for detailed description.

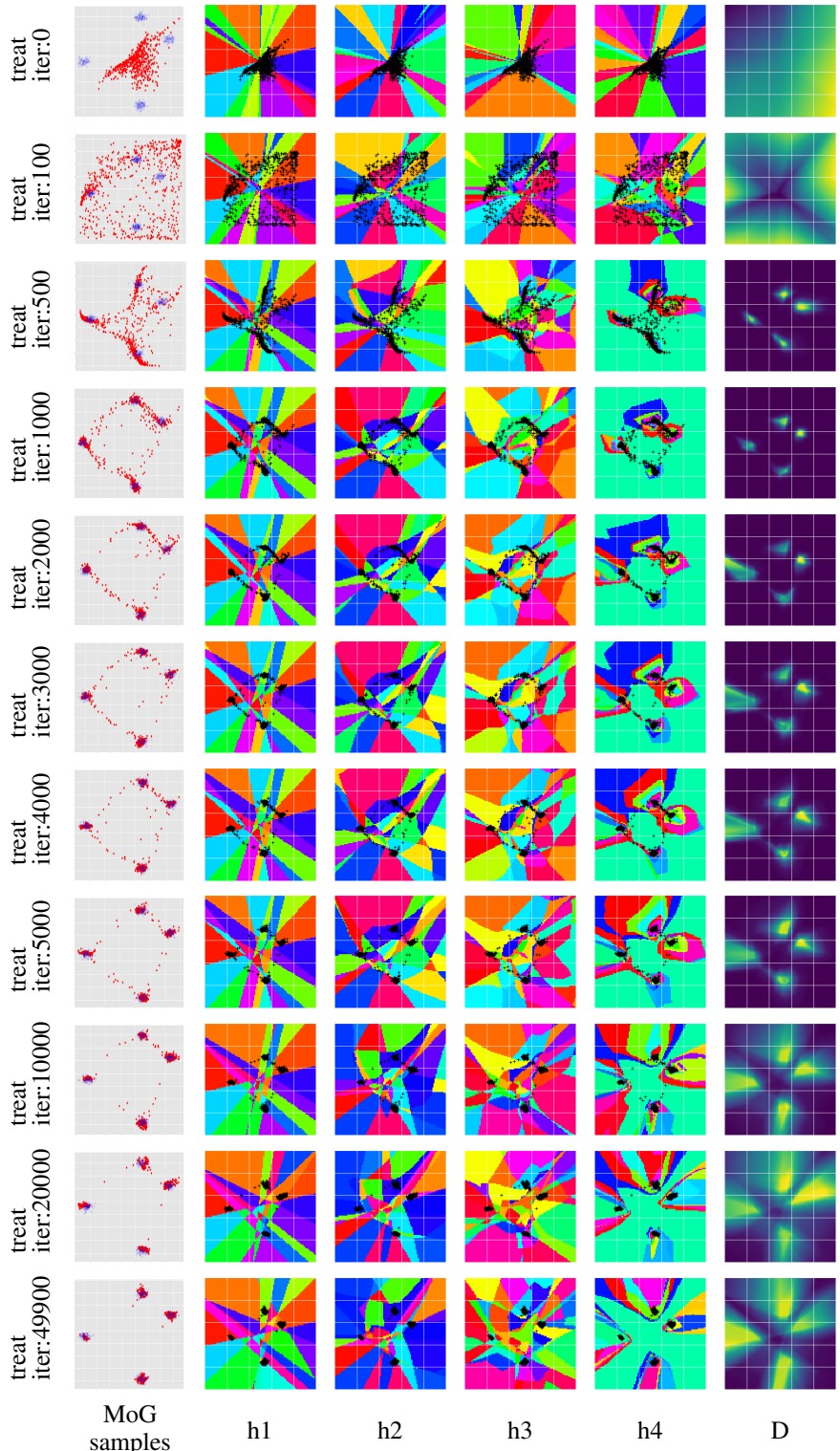

Figure 13: More Results on Fitting 2D Mixture of Gaussian on the treat group. See Figure 4 for detailed description.

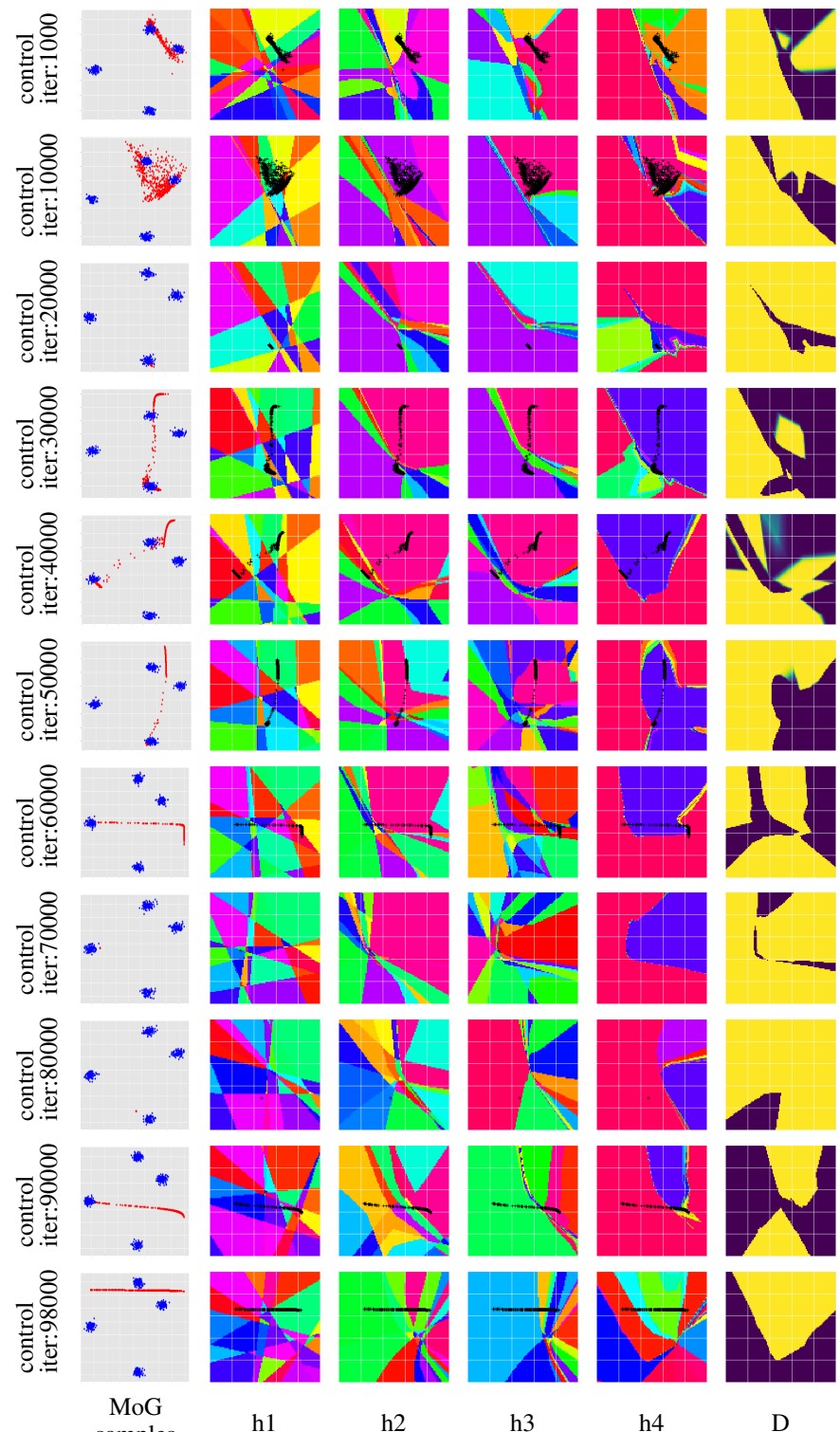

Figure 14: More Results on Fitting 2D Mixture of Gaussian on the control group. See Figure 4 for detailed description.

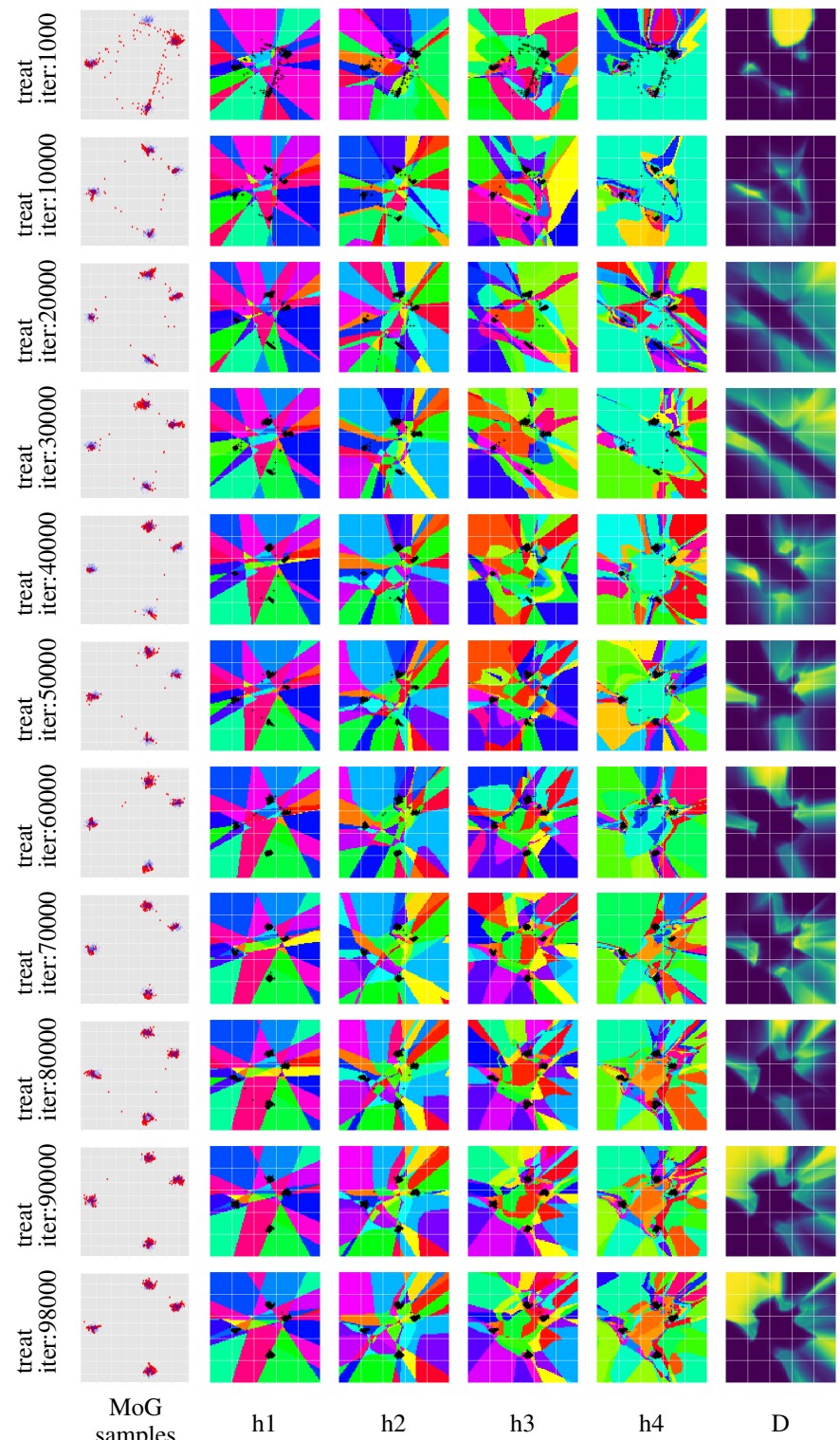

Figure 15: More Results on Fitting 2D Mixture of Gaussian on the treat group. See Figure 4 for detailed description.

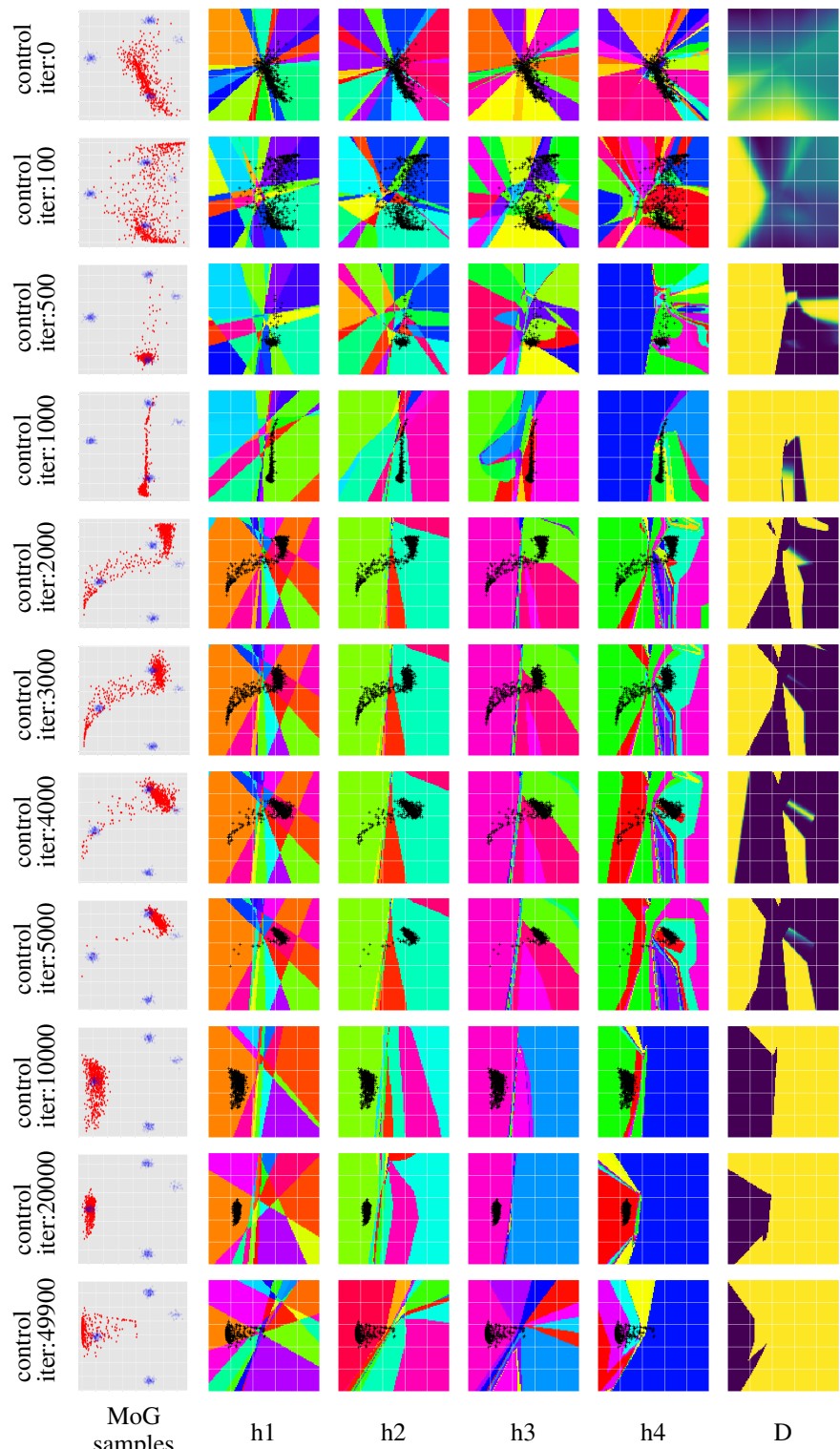

Figure 16: More Results on Fitting imbalanced 2D Mixture of Gaussian (probabilities $[.1, .3, .3, .3]$) on the control group. See Figure 4 for detailed description.

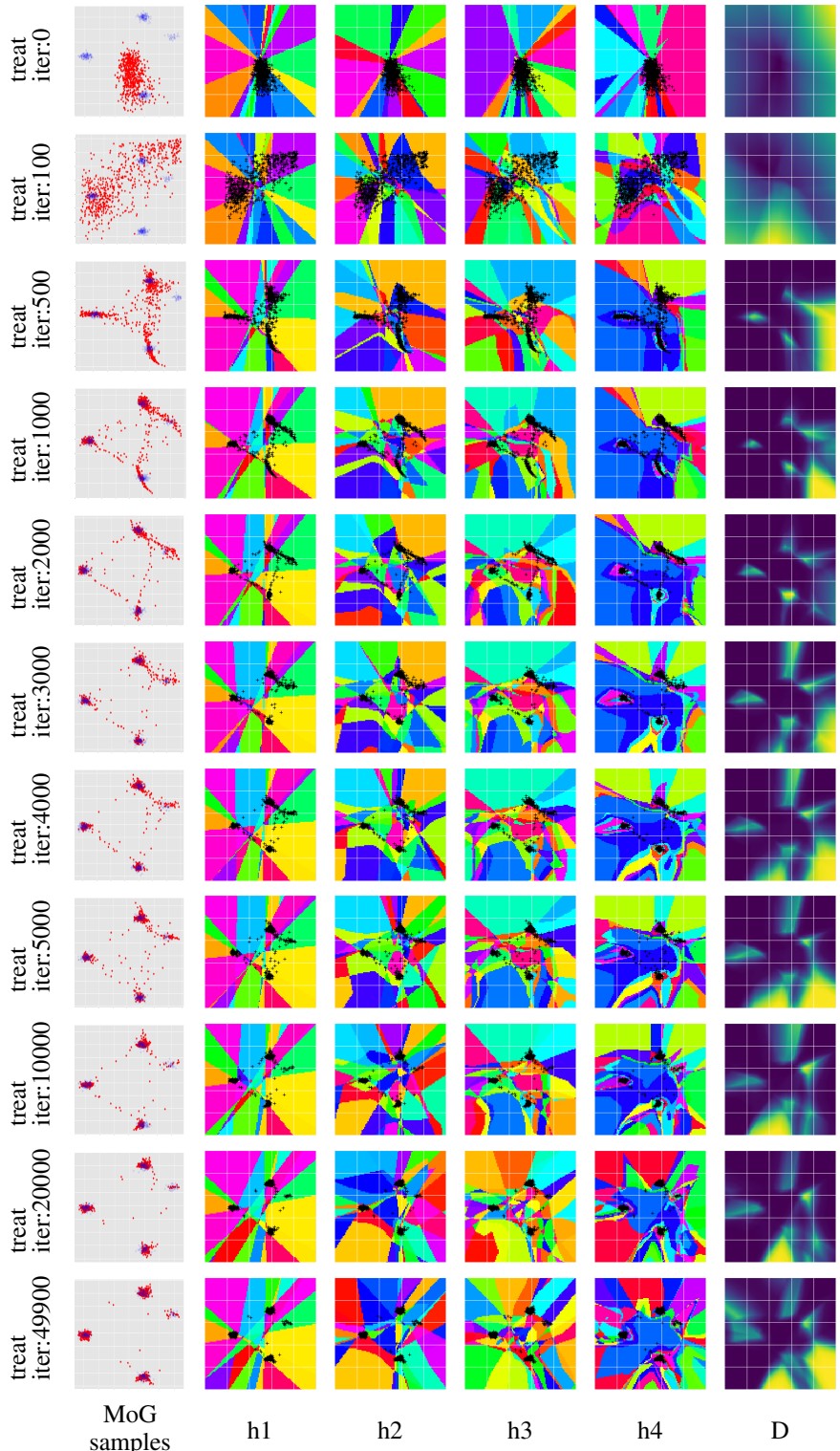

Figure 17: More Results on Fitting imbalanced 2D Mixture of Gaussian (probabilities $[.1, .3, .3, .3]$) on the treat group. See Figure 4 for detailed description.

