# OpenReview forum: "Improving GAN Training via Binarized Representation Entropy (BRE) Regularization"
_ICLR.cc/2018/Conference — Accept (Poster)_

### Official Review · AnonReviewer2 · 2017-11-27
**a cute hack with unclear impact**

**Rating:** 6
**Confidence:** 3

**Review:**

The paper proposes a regularizer that encourages a GAN discriminator to focus its capacity in the region around the manifolds of real and generated data points, even when it would be easy to discriminate between these manifolds using only a fraction of its capacity, so that the discriminator provides a more informative signal to the generator. The regularizer rewards high entropy in the signs of discriminator activations. Experiments show that this helps to prevent mode collapse on synthetic Gaussian mixture data and improves Inception scores on CIFAR10.

The high-level idea of guiding model capacity by rewarding high-entropy activations  is interesting and novel to my knowledge (though I am not an expert in this space). Figure `1 is a fantastic illustration that presents the core idea very clearly. That said I found the intuitive story a little bit difficult to follow -- it's true that in Figure 1b the discriminator won't communicate the detailed structure of the data manifold to the generator, but it's not clear why this would be a problem -- the gradients should still pull the generator *towards* the manifold of real data, and as this happens and the manifolds begin to overlap, the discriminator will naturally be forced to allocate its capacity towards finer-grained details. Is the implicit assumption that for real, high-dimensional data the generator and data manifolds will *never* overlap? But in that case much of the theoretical story goes out the window. I'd also appreciate further discussion of the relationship of this approach to Wasserstein GANs, which also attempt to provide a clearer training gradient when the data and generator manifolds do not overlap.

More generally I'd like to better understand what effect we'd expect this regularizer to have. It appears to be motivated by improving training dynamics, which is understandably a significant concern. Does it also change the location of the Nash equilibria? (or equivalently, the optimal generator under the density-ratio-estimator interpretation of discriminators proposed by https://arxiv.org/abs/1610.03483). I'd expect that it would but the effects of this changed objective are not discussed in the paper.

 The experimental results seem promising, although not earthshattering. I would have appreciated a comparison to other methods for guiding discriminator representation capacity, e.g. autoencoding (I'd also imagine that learning an inference network (e.g. BiGAN) might serve as a useful auxiliary task?).

Overall this feels like an cute hack, supported by plausible intuition but without deep theory or compelling results on real tasks (yet). As such I'd rate it as borderline; though perhaps interesting enough to be worth presenting and discussing.

A final note: this paper was difficult to read due to many grammatical errors and unclear or misleading constructions, as well as missing citations (e.g. sec 2.1). From the second paragraph alone:
"impede their wider applications in new data domain" -> domains
"extreme collapse and heavily oscillation" -> heavy oscillation
"modes of real data distribution" -> modes of the real data distribution
"while D fails to exploit the failure to provide better training signal to G" -> should be "this failure" to refer to the previously-described generator mode collapse, or rewrite entirely
"even when they are their Jensen-Shannon divergence" -> even when their Jensen-Shannon divergence
 I'm sympathetic to the authors who are presumably non-native English speakers; many good papers contain mistakes, but in my opinion the level in this paper goes beyond what is appropriate for published work. I encourage the authors to have the work proofread by a native speaker; clearer writing will ultimately increase the reach and impact of the paper.

---

> ### Author Response · Authors · 2018-01-05
> **To Reviewer2**
>
> Thank you for your detailed feedbacks and suggestions.
>
> We have made improvements in the presentation of the paper in its new version. In particular Sec. 2 is rewritten to discuss the effects of the regularizer with better clarity. We also have more compelling and comprehensive experimental results supporting the intuition. Please see the summary of change about the updates in the experiment section.
>
> To address your specific questions/comments:
>  1. “ it's true that in Figure 1b the discriminator won't communicate the detailed structure of the data manifold to the generator, but it's not clear why this would be a problem -- […]"
>
>   Ideally, when GAN training is stable, the min-max game eventually forces D to represent subtle variations in the real data distribution and passes the information to G. But when the internal representation of D is degenerate, two problems happen: 1) G under-explores, as all training signals from D could be co-linear if the fake points are in one linear region. It is unclear if G can always recover from collapsed mass caused by this. It is much more desirable for G to better explore the space from the beginning. And even if G could recover, the convergence is slowed down due to the need to correct initial mistakes. 2) large linear regions could cause the learning of G to bluntly extrapolate, resulting in large updates, which in turn drop already discovered real data modes and/or lead to oscillations.  Both of these intuitions are captured in the updated 2D synthetic plots, as well as more detailed frames in Fig. 12 and 13. Furthermore, our updated results on the convergence speed for DCGAN confirms that improved initial exploration makes convergence faster even if the dynamics were stable already.
>
>
> 2. “location of Nash equilibrium”
>
>   The locations of the Nash equilibria would change. Since D is assigned a different reward objective, there is no reason to believe that D would still have the same values for the Nash equilibria. Annealing the coefficients of the regularizer may be able to maintain these locations (which depends on the uniform convergence property of the problem and the annealing strategy, which is beyond the scope of this paper.). Our preliminary studies for annealing the regularization coefficients produced marginally inferior Inception score. We have not explored different annealing strategies yet in the experiments.
>
> 3 “discussion and comparison to Wasserstein GAN”
>
>   Thank you for the suggestion. We’ve added a discussion in Sec.\ 2.1 on other WGAN-GP and other methods that regularize gradient norm. We’ve also added comprehesive comparison in the experiments (Table 1.) showing that BRE outperforms WGAN-GP on all architectures tested.
>
> 4. “auxiliary tasks”
>
>   Indeed certain auxiliary tasks can regularize GAN training. For example, predicting image classes in semi-supervised learning GAN. BRE regularizer is compatible with semi-supervised GAN as well, and as shown in Sec. 4.3, can further improve the results.
>
>   We tested out reconstructing real data as auxiliary task to regularize D, and found that it worsens results consistently. A brief discussion is added in Sec. 5 (DISCUSSION AND FUTURE WORK), and results are shown in a table in the Appendix. We believe this is an interesting direction worth further experimentations and analysis in future work. There are a few other GAN works that use auto-encoder, such as Energy-Based GAN (Zhao et al., 2016) and Boundary Equilibrium GAN (Berthelot et al., 2017), or learning an inference network as the reviewer suggested, (Donahue et al., 2016; Dumoulin et al., 2016). It is unclear if their benefits stem from the regularization effects or the fact that other parts of GAN  (such as the objective) are modified. We added a disccussion about this in Sec. 2.1.

---

### Official Review · AnonReviewer1 · 2017-11-28
**An interesting novel regularizer for rectifier discriminators in GAN**

**Rating:** 7
**Confidence:** 4

**Review:**

The paper proposed a novel regularizer that is to be applied to the (rectifier) discriminators in GAN in order to encourage a better allocation of the "model capacity" of the discriminators over the (potentially multi-modal) generated / real data points, which might in turn helps with learning a more faithful generator.

The paper is in general very well written, with intuitions and technical details well explained and empirical studies carefully designed and executed.

Some detailed comments / questions:

1. It seems the concept of "binarized activation patterns", which the proposed regularizer is designed upon, is closely coupled with rectifier nets. I would therefore suggest the authors to highlight this assumption / constraint more clearly e.g. in the abstract.

2. In order for the paper to be more self-contained, maybe list at least once the formula for "rectifier net" (sth. like "a^T max(0, wx + b) + c") ? This might also help the readers better understand where the polytopes in Figure 1 come from.

3. In section 3.1, when presenting random variables (U_1, ..., U_d), I find the word "Bernourlli" a bit misleading because typically people would expect U_i to take values from {0, 1} whereas here you assume {-1, +1}. This can be made clear with just one sentence yet would greatly help with clearing away confusions for subsequent derivations.
Also, "K" is already used to denote the mini-batch size, so it's a slight abuse to reuse "k" to denote the "kth marginal".

4. In section 3.2, it may be clearer to explicitly point out the use of the "3-sigma" rule for Gaussian distributions here. But I don't find it justified anywhere why "leave 99.7% of i, j pairs unpenalized" is sth. to be sought for here?

5. In section 3.3, when presenting Corollary 3.3 of Gavinsky & Pudlak (2015), "n" abruptly appears without proper introduction / context.

6. For the empirical study with 2D MoG, would an imbalanced mixture make it harder for the BRE-regularized GAN to escape from modal collapse?

7. Figure 3 is missing the sub-labels (a), (b), (c), (d).

---

> ### Author Response · Authors · 2018-01-05
> **To Reviewer1**
>
> Thank you for your insightful suggestions.
>
> We have made improvements to the presentation of the paper according to the comments.
>
> “For the empirical study with 2D MoG, would an imbalanced mixture make it harder for the BRE-regularized GAN to escape from modal collapse?”
>
> Thank you for the suggestion. We have added one more set of results for imbalanced mixture distributions in the appendix of the revised paper. We find that on imbalanced mixture distributions, BRE-regularized GAN can still discover the support of infrequent modes most of the time, however, sometimes the probability mass assigned to those modes is not correct (usually under represented).

---

### Official Review · AnonReviewer3 · 2017-12-02
**Improving GANs by promoting more informative weights in hidden layers**

**Rating:** 4
**Confidence:** 3

**Review:**

The paper presents a method for improving the diversity of Generative Adversarial Network (GAN) by promoting the Gnet's weights to be as informative as possible. This is achieved by penalizing the correlation between responses of hidden nodes and promoting low entropy intra node. Numerical experiments that demonstrate the diversity increment on the generated samples are shown.

Concerns.

The paper is hard do tear and it is deficit to identify the precise contribution of the authors. Such contribution can, in my opinion, be summarized  in a potential of the form

with

$$
R_BRE = a R_ME+ b R_AC = a \sum_k  \sum_i s_{ki}^2   +  b \sum_{<k,l>} \sum_i \{ s_{ki} s_{li} \}
$$
(Note that my version of R_ME is different to the one proposed by the authors, but it could have the same effect)

Where a and b are parameters that weight the relative contribution of each term  (maybe computed as suggested in the paper).

In this formulation:

Then R_ME has a high response if the node has saturated responses -1’s or 1``s, as one desire such saturated responses, a should be negative.

The R_AC, penalizes correlation between responses of different nodes.

The point is,

a) The second term will introduce  low correlation in saturated vectors, then the will be informative.

b) why the authors use the softsign instead the tanh:  $tahnh \in C^2 $! Meanwhile the derivative id softsign is discontinuous.

c)  It is not clear is the softsign is used besides the activation function: In page 5 is said “R_BRE can be applied on ant rectified layer before the nolinearity” . This seems tt the authors propose to add a second activation function (the softsign), why not use the one is in teh layer?

d) The authors found hard to regularize the gradient $\nabla_x D(x)$, even they tray tanh and cosine based activations. It seems that effectively, the  introduce their additional softsign in the process.

e) En the definition of R_AC, I denoted by <k,l> the pair of nodes (k \ne l). However, I think that it should be for pair in the same layer. It is not clear in the paper.

f) It is supposed that the L_1 regularization motes the weights to be informative, this work is doing something similar. How is it compared  the L_1 regularization vs. the proposal?

Recommendation
I tried to read the paper several times and I accept that it was very hard to me. The most difficult part is the lack of precision on the maths, it is hard to figure out what the authors contribution indeed are. I think there is some merit in the work. However, it is not very well organized and many points are not defined. In my opinion, the paper is in a preliminary stage and should be refined. I recommend a “SOFT” REJECT

---

> ### Author Response · Authors · 2018-01-05
> **To Reviewer3**
>
> Thank you for your comments. We have improved the explanation about the motivation of this regularizer, and the math presentation of its formal definition in the revised paper. We believe that there are a few misunderstandings of our method, and we will clarify them and address the reviewer’s questions below.
>
> **Concerning the use of softsign: **
>
> “(b) why the authors use the softsign instead the tanh:  $tanh \in C^2 $! Meanwhile the derivative id softsign is discontinuous.”
> Using the softsign function to replace the sign function is to prevent null computation when $$h=0$$. Theoretically tanh with high temperature could also work. In the revised paper, we have also included the experimental results when using tanh, which shows decreased effectiveness in regularizing the GAN training comparing to softsign.  We believe the reason why our version softsign is better empirically than tanh when used in BRE is due to the scale-invariance achieved by the adaptive \epsilon. This is discussed at the end of the first paragraph in Sec. 3.3.
> Also, the derivative of the softsign function is continuous (although its 2nd order derivative is not continuous at a single point, but does not really matter for SGD).
>
> “c)  It is not clear is the softsign is used besides the activation function: In page 5 is said “R_BRE can be applied on ant rectified layer before the nolinearity” . This seems tt the authors propose to add a second activation function (the softsign), why not use the one is in teh layer?”
> No, we compute the value of R_BRE from the immediate pre-nonlinearity layer, and add this value to the objective function. The nonlinearity of the networks are not changed. We have added a figure (Figure 2) in the revised paper to clarify this.
>
> “d) The authors found hard to regularize the gradient $\nabla_x D(x)$, even they tray tanh and cosine based activations. It seems that effectively, the  introduce their additional softsign in the process.”
> Regularizing the diversity of $\nabla_x D(x)$ is a straightforward naive approach if we want rich diverse training signal for G. However, this does not work for rectifier nets, for reason analysed in Sec. 5 (Discussion), and hence one of the significance of our contribution. The use of softsign is unrelated to this issue.

---

> > ### Author Response · Authors · 2018-01-05
> > **To Reviewer3 continued**
> >
> > ** Reviewer’s interpretation of the regularizer and potential confusion about its effect: **
> >
> > “ in my opinion, be summarized in a potential of the form with
> > $$R_{BRE} = a R_{ME}+ b R_{AC} = a \sum_k  \sum_i s_{ki}^2   +  b \sum_{<k,l>} \sum_i \{ s_{ki} s_{li} \}$$
> > “
> >
> > Our formulation of BRE is:
> > $$R_{ME} = \frac{1}{d}\sum_{k=1}^d \bar{s}_{(k)}^2 = \frac{1}{d}\sum_{k=1}^d \frac{1}{K^2}(\sum_{i=1}^{K}s_{k,i})^2$$
> > $$R_{AC} =  \text{avg}_{i \neq j}  | s_{i}^{T}s_{j}| / d = \frac{1}{K(K-1)} \sum_{i\neq j} | \sum_{l=1}^d s_{il}s_{jl}| / d.$$
> >
> > Let RRME denote the RME term proposed in the reviewer’s comment, and RME be this term in the paper. Similarly let RRAC and RAC denote the RAC term in the comment and in the paper, respectively. The motivation of our regularizer is to encourage a large entropy of the activation vector on some particular layer of D, so that D would provide informative learning signals for G.
> >
> > The activation vector s is a binary vector (each element is either +1 or -1), computed from a sign function. So $$s_{k,i}^2$$ will always be 1 in RRME and this term will be ineffective. On the other hand, RME encourages $$\sum_{i=1}^{K}s_{k,i}$$ to be 0, i.e. zero mean for the k-th hidden unit. We would like to encourage this zero-mean property for the purpose of increasing its entropy (and thus $$a$$ is set to be positive).
> >
> > For the second term, note that RAC has an absolute value on the correlation of s_i and s_j (they are approximately zero-mean because of RME), which encourage s_i and s_j to be independent. Both positive correlation and negative correlation are penalized. Thus again RAC encourage a large entropy.  On the other hand, penalizing RRAC is actually encouraging s_k and s_l to be negatively correlated. The minimal value of the inner product of s_k and s_l is -d. However, it is impossible for all the pairs of (s_k, s_l) to simultaneously achieve this minimal value. It seems not easy to analyze when (s_1, …, s_K) would achieve its minimal value on RRAC, which makes it difficult to interpret its effect.
> > Overall, we don’t see that the proposed regularizer in the review has a similar effect as the BRE regularizer proposed in the paper.
> >
> > “e) En the definition of R_AC, I denoted by <k,l> the pair of nodes (k \ne l). However, I think that it should be for pair in the same layer. It is not clear in the paper.”
> > Yes, that is right. The summation is over all the pairs in the same layer. We have added  a footnote for the definition of R_BRE to clarify this point in the new version of the paper.
> >
> > “f) It is supposed that the L_1 regularization motes the weights to be informative, this work is doing something similar. How is it compared  the L_1 regularization vs. the proposal?”
> > Our method is to encourage D to have diverse activation patterns, so that G could have more informative signals for learning. The BRE regularizer is computed based on the pre-nonlinearity value of the hidden nodes, while L_1 regularization is applied to the weights of D. We don’t see the connection between the L_1 regularization and the BRE regularizer at the moment.

---

### Author Response · Authors · 2018-01-05
**Summary of changes**

We thank all reviewers for their detailed feedbacks and insights. Following remarks by the reviewers, we significantly revised the writing of the paper. The flow and the structure are mostly the same, but the language is changed significantly in most paragraphs to achieve better clarity hopefully.  Illustrations are also added in Section 3 to help to interpret the notations.  We would also like to shorten the title to “Improving GAN Training via Binarized Representation Entropy Regularization” if this paper is accepted.

The experiment section still demonstrates the same claims, but now with more comprehensive comparisons and some more compelling new results.  In particular, Table 1 shows improvements over both baselines without the regularizer and WGAN-GP, on four different architectures. Figure 5 shows that for DCGAN that is already well-engineered to be stable, adding the regularizer makes the convergence to equilibrium significantly faster (shown by repeated runs with error bars). In the semi-supervised learning setting, we now report a systematic comparison from 10 random runs in Table 2, rather than curves from a single run using plots. Furthermore, we report semi-supervised classification results on SVHN which were not included in the original draft. The new results on SVHN show that even in semi-supervised learning with feature matching, GAN training occasionally fails, depending on the random seeds. But the proposed regularizer dramatically reduces this failure rate. The CelebA experimental results are now moved to the Appendix.  A different set of 2D synthetic experimental results are shown in the main paper now (old ones are moved to the Appendix), which conveys the main ideas slightly better.
Taken together, we believe that the updated experimental results provide much stronger evidences for our claim: the proposed BRE regularizer improves exploration in the initial phase of GAN training, so that G can discover various modes/parts of the real data manifold more successfully (pictorially illustrated in Fig 1); training is more stable because there is fewer large linear regions, in which D bluntly extrapolates and G makes large jumps as results; final sample quality is better following more stable training and less mode dropping; semi-supervised classification accuracy is also improved.

Due to writing problems in the original draft, we feel that the main idea of the paper was not sufficiently laid out with clarity. This is now improved in the revised paper, and we would like to summarize it here again, briefly:

- Our motivation is the following: when D spreads out its model capacity, the evenly dispersed partitioning of the input data space helps G to explore better and discover the real data manifold, while avoiding large unstable jumps due to erroneous extrapolation made by D.

- When the classification task for D is too simple, the internal layers of D could have degenerate representations due to overfitting, whereby large portions of the input space are modelled as linear regions, as depicted in the illustration in Fig. 1 and shown by the synthetic experiment in Fig. 4 (control run without BRE), more plots in Fig. 12-17 in the appendix.  This could happen at the beginning of the training, or even in the later stages for high dimensional data.

 - With such degeneracy, learning signals from D are not diverse and G could under-explore the input space at the beginning, potentially missing out on distinct faraway modes. G might recover later, but it is more desirable to sufficiently explore at the beginning, both for a better chance of capturing all modes and for faster convergence.

- Furthermore, the large linear regions in the degenerate representation would cause the learning of G to bluntly extrapolate, producing large jumps that could drop the current modes and/or lead to oscillations. This phenomenon can be observed in the experiments on synthetic data in Figure 4, and Fig. 12-17 in the appendix.

- If the model has already been well-engineered, such as DCGAN,  the improvement by BRE is less dramatic at the end of training (still yielding some improvement, as shown by the DCGAN results in Table 1 on page 8 of the revised paper). However, the speed of convergence to equilibrium with BRE regularization is significantly faster (Figure 5 of the revised paper), thanks to the improved exploration at the beginning.

---

### Author Response · Authors · 2018-01-31
**Title change**

As mentioned in the rebuttal, we now changed the title to "Improving GAN Training via Binarized Representation Entropy (BRE) Regularization". The original one was "Improving diversity in generative adversarial networks by encouraging discriminator representation entropy".

---

### Decision · Program_Chairs · 2018-01-29
**ICLR 2018 Conference Acceptance Decision**

**Decision:**

Accept (Poster)

**Comment:**

  + Original regularizer that encourages discriminator representation entropy is shown to improve GAN training.
  + good supporting empirical validation
  - While intuitively reasonable, no compelling theory is given to justify the approach
  - The regularizer used in practice is a heap of heuristic approximations (continuous relaxation of a rough approximate measure of the joint entropy of a binarized activation vector)
  - The writing and the mathematical exposition could be clearer and more precise